# Intra- and Interspecies Conjugal Transfer of Plasmids in Gram-Negative Bacteria

**DOI:** 10.3390/biomedicines13010238

**Published:** 2025-01-20

**Authors:** Julia R. Savelieva, Daria A. Kondratieva, Maria V. Golikova

**Affiliations:** Department of Pharmacokinetics & Pharmacodynamics, Gause Institute of New Antibiotics, 11 Bolshaya Pirogovskaya Street, 119021 Moscow, Russia; savmos80@mail.ru (J.R.S.); goawaymrway@gmail.com (D.A.K.)

**Keywords:** conjugation, *Klebsiella pneumoniae*, *Escherichia coli*, *Pseudomonas aeruginosa*, meropenem, carbapenemases, hollow-fiber infection model

## Abstract

**Background/Objectives:** Plasmid-mediated resistance is a significant mechanism that contributes to the gradual decrease in the efficacy of antibiotics from various classes, including carbapenems. The aim of this study is to investigate the frequency of transfer of carbapenemase-encoding plasmids from *K. pneumoniae* to *E. coli* and *P. aeruginosa*. **Methods:** Matings were performed on agar with subsequent isolation of transconjugant, recipient, and donor colonies. The frequency of conjugation (CF) and minimum inhibitory concentrations (MICs) of meropenem were determined for the PCR-confirmed transconjugants. A pharmacodynamic study was conducted using a hollow-fiber infection model on *E. coli* transconjugant in order to evaluate its viability in the presence of therapeutic concentrations of meropenem. **Results:** CF for *K. pneumoniae*-*K. pneumoniae* was similar to that for *K. pneumoniae*-*E. coli* and was higher the higher was meropenem MIC of the *K. pneumoniae* donor. The meropenem MICs for *K. pneumoniae* and *E. coli* transconjugants were higher (0.25–4 μg/mL) compared to recipients (0.03–0.06 μg/mL). *P. aeruginosa* did not acquire plasmids from *K. pneumoniae*. In pharmacodynamic experiments, an *E. coli* transconjugant with MIC of 2 mg/L within the “susceptibility range”, failed to respond to meropenem treatment. **Conclusions:** The frequency of conjugation between *K. pneumoniae* and *E. coli* falls within a similar range. A higher permissiveness of *K. pneumoniae* for plasmids from *K. pneumoniae*, i.e., within the same species, was observed. Conjugation did not occur between *K. pneumoniae* and *P. aeruginosa*. The transconjugants with meropenem MICs with borderline susceptibility may pose a potential threat to the efficacy of meropenem.

## 1. Introduction

Horizontal gene transfer, in particular via plasmids through conjugation, is a major mechanism that contributes to the spread of antibiotic-resistant bacteria around the world [1,2]. However, after nearly a century of research, there is still much to be learned about the conjugation of bacteria. It is crucial to enhance our understanding of this process, as the spread of plasmids carrying antibiotic resistance genes among different species of Gram-negative bacteria is thought to be a major cause of infections that pose a threat to global health. Previous studies have demonstrated that the efficacy of conjugation is influenced by a variety of factors, including the individual characteristics of the donor and recipient cells, the bacterial species involved [3], the levels of antimicrobial resistance, the number of plasmids present in the donor cell [4], and the conjugative permissiveness of the recipient [5].

Carbapenems, which are among the most commonly prescribed antimicrobial agents for a wide range of clinical indications, are at risk of becoming ineffective due to the emergence of resistant strains carrying plasmids with carbapenemase genes [6]. Among the bacterial species commonly found to carry carbapenemase genes encoding carbapenem resistance, *Klebsiella pneumoniae* is one of the most prevalent. Various diseases, including bloodstream infections, pneumonia, and urinary tract infections, can be caused by *K. pneumoniae* [7]. Individuals with weakened immune systems are particularly vulnerable to serious infections caused by this bacterium. Treating infections caused by these strains has become more challenging due to their increased resistance to antibiotics. The most widespread carbapenemases, including KPC, OXA-48 and NDM, are associated with this species [8,9]. The range of bacteria that can exist in the natural environment of *K. pneumoniae* and participate in conjugation processes is quite wide; one of the most likely partners for plasmid transfer is *Escherichia coli*. *K. pneumoniae* inhabits the same environment as *E. coli* and their interaction is common. *Pseudomonas aeruginosa* often occurs alongside *K. pneumoniae* and *E. coli* at the same location. In addition, *P. aeruginosa* is known to cause serious hospital-acquired infections that can be difficult to treat, particularly due to producing carbapenemases. Several studies have reported the isolation of clinical strains of *P. aeruginosa* carrying KPC carbapenemase genes [10,11].

The focus of our study was to investigate the efficiency of conjugation between various bacterial species, both closely and distantly related. We investigated *K. pneumoniae*, *E. coli*, and *P. aeruginosa*, as well as conducting a pharmacodynamic evaluation of meropenem against a transconjugant strain generated through mating experiments. The aim was to assess its viability in the presence of an antibiotic, simulating its clinical dosing regimen using a hollow-fiber infection model (HFIM). The HFIM is an effective tool for studying antimicrobial efficacy and has numerous applications, including investigating bacterial resistance and assessing the clinical efficacy of antimicrobials [12]. In this study, epithelial lining fluid (ELF) meropenem pharmacokinetics [13] was simulated following the administration of 2 g every 8 h in a 3 h infusion during 5-day treatments in an HFIM.

The investigation of issues related to the conjugation efficiency between clinically relevant strains of Gram-negative bacteria, assessing their relative potential to acquire plasmids with carbapenemase genes, and subsequently replicating them in order to survive under meropenem exposure, determines the clinical significance of this study.

## 2. Materials and Methods

### 2.1. Antimicrobial Agents, Bacterial Strains and Susceptibility Testing

Meropenem and CCCP (3-chlorophenylphenhydrazone) powders were purchased from Sigma-Aldrich (St. Louis, MO, USA). Three clinical isolates of *K. pneumoniae* were used as plasmid donors in mating experiments: 38, 485, and 565 (Table 1).

Meropenem-susceptible carbapenemase-non-producing strains of *E. coli* K-12 C600, *E. coli* ATCC 25922, *K. pneumoniae* ATCC 700603, *K. pneumoniae* 188 (clinical isolate), *P. aeruginosa* ATCC 9027, and *P. aeruginosa* ATCC 27853 were used as recipients in mating experiments. Plasmid donors [14,15] and the recipient strains *K. pneumoniae* 188 and *E. coli* C600 were kindly provided by Dr. Ageevets A.V., Pediatric Research and Clinical Center for Infectious Diseases, Saint Petersburg, Russia.

Susceptibility testing was carried out using broth microdilution techniques with a standard inoculum of approximately 5 × 10^5^ CFU/mL. Meropenem MICs were determined according to standard recommendations using cation-supplemented Mueller–Hinton broth (CSMHB) (Becton Dickinson, Franklin Lakes, NJ, USA) [16]. Before reading, microplates were incubated at 37 °C for 18–20 h. MIC values in each case were obtained at least in triplicate, and modal MICs were estimated.

The susceptibility of *P. aeruginosa* isolates to meropenem was also tested in the presence of the efflux pump inhibitor CCCP (3-chlorophenylphenhydrazone) at a concentration of 50 μg/mL using the standard susceptibility testing technique. The efflux pump inhibitor solution was freshly prepared before every experiment and every test was carried out on a 96-well plate. Additionally, it has been verified that the growth of all *P. aeruginosa* strains is not inhibited in the presence of 50 µg/mL of an inhibitor. Resistance due to the efflux pumps was presumed if the MIC of the meropenem decreased by 4-fold or more in the presence of the inhibitor [17].

### 2.2. Mating Experiments

The protocol of mating experiments followed to obtain transconjugants is summarized in Figure 1 [18].

To distinguish between donor, recipient and transconjugant cells, all the recipient strains (*E. coli* C600, *E. coli* ATCC 25922, *K. pneumoniae* ATCC 700603, *K. pneumoniae* 188, *P. aeruginosa* ATCC 9027, and *P. aeruginosa* ATCC 27853) were incubated on the media (three passages) with rifampicin (at concentrations from 50 to 250 µg/mL) to produce rifampicin-resistant mutants. These mutants have MICs of rifampicin greater than 512 µg/mL. For all experiments, these rifampicin-resistant mutants were used.

Bacteria were grown in Luria broth (LB, Becton Dickinson, Franklin Lakes, NJ, USA) and Luria agar (LA, Becton Dickinson, Franklin Lakes, NJ, USA) media at 37 °C. When required, LA was supplemented with antimicrobial agents at the following final concentrations (µg/mL): meropenem (0.5–2) and rifampicin (150–200).

Matings were performed overnight on the LA plates according to a previously published protocol with minor modifications [18]. Briefly, the 1:1 mixture of the donor and recipient in the late logarithmic growth phase was plated on the LA surface and incubated at 37 °C for 18–20 h. The mixed growth was then scraped from the plate surface and resuspended in 1 mL of saline, and to quantify the numbers of donor, recipient, and transconjugant cells, the cell mixture was diluted as appropriate. Subsequently, 100 µL samples were spread on appropriate selective plates with meropenem and/or rifampicin at the following final concentrations: meropenem (equal to 4×MIC of recipient) and rifampicin (150–200 µg/mL). Parent strains were plated in parallel with the matings and then processed similarly to the matings as controls. Isolated colonies from matings presumed as transconjugant and parental strains from controls were used to identify recombinants and parental forms. The conjugation efficiency was assessed by the conjugation frequency (CF): ratio of the number of CFU of transconjugants per mL to the number of CFU of recipients plus transconjugants per mL [19]. The data corresponding to the absence of conjugation as confirmed by the PCR are depicted as equivalent to a limit of detection for conjugation frequency of 10⁻⁹.

“Possible” transconjugants were screened by streaking colonies from the selection plate with meropenem and rifampicin onto a fresh plate with the same antibiotics to look for growth. The plasmid acquisition by recipient strains was confirmed by the PCR with primers specific to genes encoding plasmid replication proteins and relaxases (Appendix A). In addition, a PCR was performed to confirm or rule out the presence of carbapenemase genes in potential transconjugant strains. A PCR was performed according to a standard protocol for amplification of fragments with a size of 1 kb [20].

### 2.3. In Vitro Dynamic Model and Operational Procedure Used in the Pharmacodynamic Experiments

The HFIM was used to evaluate meropenem pharmacodynamics and to conduct growth control experiments. The flowchart for the pharmacodynamic experiments is presented in Figure 2.

The studies were performed using a hollow-fiber bioreactor (Fresenius dialyzer, model AV400S, Fresenius Medical Care AG, Bad Homburg, Germany) that represents the infection site (HFIM schematic illustration is provided in Appendix A). The operational procedure is described in detail elsewhere [21]. Briefly, antibiotic dosing and sampling were processed automatically, using computer-assisted controls. The system was filled with sterile CSMHB and placed in an incubator at 37 °C. The inoculum of an 18 h culture of *E. coli* was injected into the hollow-fiber bioreactor to produce a bacterial concentration of 10^8^ CFU/mL. After a 2 h incubation, samples were obtained to determine the starting bacterial concentration; then, the infusion of CSMHB with antibiotic was initiated. The duration of each experiment was 120 h. To verify the reliability of pharmacokinetic simulations, throughout each experiment the bioreactor was sampled immediately after the end of infusion (3 h) and at the 6th hour of the dosing interval.

In each experiment, the bacteria-containing medium from the central unit of the model was sampled to determine bacterial concentrations throughout the observation period. Samples (100 µL) were serially diluted as appropriate and 100 µL was plated onto Mueller–Hinton agar plates, which were placed in an incubator at 37 °C for 24 h. The lower limit of accurate detection was 1 × 10^2^ CFU/mL (equivalent to 10 colonies per plate).

To monitor the time courses of antibiotic-resistant subpopulations of *K. pneumoniae* in the pharmacodynamic experiments, the central unit of the model was multiply sampled throughout the observation period (120 h). The samples were serially diluted, if necessary, plated on Mueller–Hinton agar (MHA) with 2×, 4×, 8× and 16×MIC of meropenem, and incubated for 24–48 h at 37 °C. The viable counts were screened visually for growth. The lower limit of detection was 10 CFU/mL (equivalent to at least one colony per plate).

### 2.4. Antibiotic Dosing Regimens and Simulated Pharmacokinetic Profiles

Meropenem treatment mimicked the therapeutic dosing regimen: 2000 g administered every 8 h, as a 3 h intravenous infusion. A mono-exponential profile in epithelial lining fluid (ELF) after thrice-daily dosing of meropenem with a half-life (*t*1/2) of 1.4 h was simulated [13] for five consecutive days with a total of 15 infusions. The pharmacokinetic parameter values were as follows: C_MAX_ = 32.4 µg/mL, 24-h area under the concentration–time curve (AUC) = 375 (µg × h)/mL. Before all pharmacodynamic simulations, the system was calibrated and preliminary in vitro pharmacokinetic experiments in CSMHB without bacteria were conducted.

### 2.5. Statistical Analysis

The reported MIC data were obtained by calculation of the respective modal values. The resultant conjugation frequency was calculated as arithmetic mean ± standard deviations for three replicate experiments. The data from each group were analyzed for statistically significant differences (*p* < 0.05) in the data mean values between the groups using SigmaPlot 12 statistical software (Systat Software Inc., headquartered in San Jose, CA, USA) by a paired two-sample *t*-test.

In pharmacodynamic and growth control experiments, bacterial count data were calculated as arithmetic mean ± standard deviations for three replicate experiments. Based on these data, kinetic growth and time-kill curves were constructed.

## 3. Results

### 3.1. Donor and Recipient Strains, Meropenem and Rifampicin Susceptibility

In mating experiments, three *K. pneumoniae* donor strains were used. These strains carried plasmids with carbapenemase genes (*bla*_KPC-2_ or *bla*_OXA-48_) and varied in resistance to meropenem. As recipients, meropenem-susceptible bacterial strains from different species that do not produce carbapenemases were used. These included closely related strains of *E. coli* and *K. pneumoniae*, both members of the *Enterobacteriaceae* family, and *P. aeruginosa*. Table 2 presents the MICs of meropenem for bacterial strains tested.

### 3.2. Mating Experiments and Meropenem Susceptibility of Transconjugants

The flowchart in Figure 3 illustrates the data on paired strains used in the mating experiments and the meropenem susceptibility of the resulting transconjugant strains.

As described in the Materials and Methods section, confirmation of the transconjugants was performed by plating different transconjugant colonies on agar plates supplemented with meropenem and rifampicin and by performing PCR analysis. *E. coli* transconjugant colonies growing on agar plates containing meropenem (plus rifampicin), regardless of the plasmid donor, were difficult to cultivate in the presence of meropenem due to the fitness cost. Only a few samples maintained steady growth after repeated cultivation. PCR analysis of *E. coli* transconjugants confirmed plasmid carriage in only 25% of the samples (Appendix A). Only these colonies were used to calculate the conjugation frequencies in *K. pneumoniae*-*E. coli* pairs. These colonies exhibited good growth on media with or without meropenem and retained their plasmids during 15 passages. In contrast, *K. pneumoniae* transconjugants showed steady growth when cultivated on agar with meropenem (plus rifampicin), and plasmid carriage was confirmed in 100% of the samples (during 15 passages).

We should pay special attention to the results obtained from the mating of *K. pneumoniae* with *P. aeruginosa*. The “possible” transconjugants of *P. aeruginosa* seemed to grow well on agar with meropenem (plus rifampicin) and were further successfully cultivated in the presence of antibiotic. However, according to PCR analysis, the percentage of plasmid carriage among these colonies was equal to 0% (Appendix A). Given the potential for *P. aeruginosa* to transform carbapenemase genes from plasmids and use them to resist meropenem, we conducted PCR analysis to detect the presence of specific genes in these isolates. The primers used to detect carbapenemase genes are presented in Appendix A of the Appendix A. As a result, PCR confirmed the absence of *bla*_KPC-2_ and *bla*_OXA-48_ genes in potential transconjugants, indicating that transformation had not occurred, and these strains relied on their intrinsic ability to resist meropenem, using efflux pumps, for example. To verify this assumption, we determined meropenem MICs for all *P. aeruginosa* isolates in the absence and presence of an efflux pump inhibitor, CCCP (3-chlorophenylphenhydrazone) (Table 3). Based on the results obtained, it was found that resistance to meropenem in three of six isolates is due to the functioning of efflux pumps. However, other *P. aeruginosa* isolates did not show a decrease in the meropenem MIC in the presence of CCCP.

All transconjugants from matings of *K. pneumoniae* with either *K. pneumoniae* or *E. coli*, which appeared to be steadily growing on agar with meropenem and were confirmed to carry plasmids, were then tested for meropenem susceptibility. As shown in Figure 3, transconjugants of *K. pneumoniae* and *E. coli*, isolated after mating with the donor *K. pneumoniae* 565 (carrying a plasmid with the *bla*_KPC-2_ carbapenemase genes), had higher meropenem MICs compared to those isolated after mating with donors *K. pneumoniae* 485 and 38 (both carrying plasmids with the *bla*_OXA-48_ carbapenemase genes). Specifically, in the first case (donor *K. pneumoniae* 565) for *K. pneumoniae* transconjugants meropenem MICs were 4 µg/mL (up to a 132-fold MIC increase compared to the recipient), while in the second case (donors *K. pneumoniae* 485 and 38) MICs were 0.25 and 0.5 µg/mL (up to a 16-fold MIC increase compared to the recipient). Similarly, in the first case (donor *K. pneumoniae* 565) for *E. coli* transconjugants meropenem MICs were 1–2 µg/mL (up to a 64-fold MIC increase compared to the recipient) while in the second (donors *K. pneumoniae* 485 and 38)-0.25 and 0.5 µg/mL.

With *P. aeruginosa*, unlike the isolated colonies did not carry plasmids, we evaluated the meropenem MICs for them as they could grow on the agar with meropenem. As shown in the flowchart, these isolates appeared to have meropenem MICs of 4–8 µg/mL.

In order to provide a more comprehensive characterization of isolated transconjugant strains of *K. pneumoniae* and *E. coli*, and to gain an understanding of how their susceptibility profiles had changed following conjugation, we assessed each isolate for the expression of plasmid-encoded resistance elements (Appendix A). As can be seen from the Tables, resistance to antibiotics in transconjugants compared to recipients is caused by genes located on plasmids. In the case of pOXAAPSS1/2 plasmids, only resistance to beta-lactams was transferred to initially susceptible recipient strains, while in the case of pKPCAPSS, resistance to both beta-lactam and fluoroquinolones was transferred, as well as resistance to macrolides.

### 3.3. Conjugation Frequency

The results of matings were analyzed by determining the conjugation frequency as the ratio of number of transconjugants confirmed by PCR to the number of recipients (data summarized in Appendix A). In general, the conjugation frequency was similar between *K. pneumoniae* and *K. pneumoniae* or *E. coli* and varied from 10^−7^ to 10^−3^ or from 10^−7^ to 10^−4^, respectively. In order to investigate specific aspects of the plasmid transfer between donors and recipients, we generated a histogram, which is illustrated in Figure 4. We compared the frequency of conjugation between the donor strains of *K. pneumoniae* and the recipient groups. As can be seen from the graph, for the recipient *E. coli* and *K. pneumoniae* strains, the conjugation rates differ depending on the donor strain of *K. pneumoniae*. When *K. pneumoniae* 38 acted as the donor, the conjugation frequency (of ~10^−7^) was relatively lower than that for *K. pneumoniae* 565 (ranging from ~10^−5^ to ~10^−4^) and *K. pneumoniae* 485 (of ~10^−6^) donors. The conjugation frequency increased depending on the donor in the following order: *K. pneumoniae* 38 < 485 < 565. Therefore, for each recipient strain, the highest rate of plasmid transfer was observed when the donor strain was *K. pneumoniae* 565. The rate was then higher when the donor was *K. pneumoniae* 485, compared to *K. pneumoniae* 38, with which the lowest rate of transfer was recorded. Among *P. aeruginosa*, no similar trends were observed, as the conjugation could not be detected. Described differences were statistically significant (*p* < 0.05).

### 3.4. Pharmacodynamic Evaluation of E. coli Transconjugant Strain

As was previously shown, transconjugants of *K. pneumoniae* and *E. coli* with increased MICs were obtained in mating experiments. The MICs were higher for *K. pneumoniae* (4 μg/mL instead of 0.03–0.06 μg/mL in the recipient) and *E. coli* (2 μg/mL instead of 0.03 μg/mL in the recipient), which carry plasmids containing *bla*_KPC–2_ carbapenemase genes.

In the context of the emergence of carbapenem-resistant bacterial strains in clinical settings and the decreasing efficacy of carbapenems, it is crucial to investigate whether meropenem monotherapy still maintains its efficacy against these bacteria, despite their ability to produce carbapenemases. In this light, the pharmacodynamic study was designed to demonstrate how transconjugant strains are able to resist meropenem, when its concentrations correspond to those found in clinical settings and are constantly changing, as occurs in humans in the infection site. In other words, this aspect of the study allows us to anticipate and understand the chance of such strains surviving after they arise from contact with a plasmid donor (using the benefits of a new plasmid containing carbapenemase genes), and subsequently being exposed to antibiotics. For this purpose, we conducted pharmacodynamic experiments with meropenem and the KPC carbapenemase-producing *E. coli* transconjugant strain C600/565, with a meropenem MIC corresponding to the “susceptibility” breakpoint of 2 µg/mL (according to EUCAST guidelines) and with the *E. coli* C600 recipient that served as a control. In these experiments, the pharmacokinetics of meropenem observed in the ELF after a high-dose regimen of 2 g every 3 h was simulated.

The results of simulations with recipient strain *E. coli* C600 and transconjugant strain *E. coli* C600/565 are shown in Figure 5.

As seen from the figure, the numbers of *E. coli* C600/565 (pKPCAPSS transconjugant) increased rapidly after an initial 6 h decline; bacterial regrowth was accompanied by the intensive selection of meropenem-resistant cells. The recipient strain *E. coli* C600 was found to be eliminated from the bioreactor within the first 6 h. That is, initially, this strain without plasmid was unable to develop meropenem resistance.

## 4. Discussion

As the result of our investigation, we detected that overall conjugation frequency in pairs *K. pneumoniae*-*K. pneumoniae*, and *K. pneumoniae*-*E. coli* varied in a wide range from high (~10^−3^) to low (~10^−7^). Apparently, plasmid transfer is dependent on a combination of internal features as well as recipient and donor strains. It is known that many factors can influence these processes, such as plasmid-encoded regulatory elements, the specificity of interaction between host chromosomes and plasmids, and the external environment [23]. In addition, conjugation dynamics may depend on both the plasmid acquisition cost and the fitness cost [24], existence of a specific recipient cell surface for conjugative transfer [25], trade-off between lag times, and growth rate [26].

When analyzing the histograms presented in Figure 4, we identified that the meropenem resistance level of the donor was consistent with the effectiveness of the horizontal gene transfer in pairs *K. pneumoniae*-*K. pneumoniae* and *K. pneumoniae*-*E. coli*. One possible explanation for the higher conjugation rates in donors with greater antibiotic resistance is that they may have a larger number of plasmids per cell compared to less resistant donors. This could make them more efficient at conjugation. However, there is also evidence to suggest an inverse relationship between the number of plasmid copies per cell and the size of the plasmid [27,28]. This information suggests that, in our study, the strain *K. pneumoniae* 565, with the largest plasmid (127,970 bp), may have fewer copies compared to other strains (*K. pneumoniae* 485 and 38), with smaller plasmids (63,359 bp and 66,284 bp, respectively) [29]. Consequently, the number of plasmid copies may not fully explain why the highest conjugation rate was observed in the donor strain, which presumably has a lower number of copies, or why the rates also differed between pairs with donors, *K. pneumoniae* strains 485 and 38, both carrying plasmids of similar sizes. Additional factors may influence the conjugation frequency, such as the carbapenemase gene doses [30] and level of their expression [31], and the carbapenem enzyme activity [32,33]. It is likely that a combination of these factors contributes to the outcome.

It is well known that carbapenem-hydrolyzing enzymes such as KPC carbapenemases are more efficient at degrading carbapenem antibiotics compared to OXA-48 carbapenemases [32,33]. Interestingly, transconjugants carrying *bla*_KPC-2_ genes (*K. pneumoniae* and *E. coli*) demonstrated higher levels of resistance to meropenem compared to transconjugants carrying *bla*_OXA-48_ carbapenemases as measured by their MICs (2–4 µg/mL versus 0.25–0.5 µg/mL) (Figure 3). This may be due to the individual plasmid properties and their acquisition cost and/or fitness cost [24] in recipient cells. It is worth noting that we observed a fitness cost in all *E. coli* transconjugants. In addition, multiple carbapenemase gene copies in the transconjugant plasmids may also contribute to the higher meropenem MICs [30].

Unlike *K. pneumoniae* and *E. coli*, *P. aeruginosa* exhibited distinctive characteristics that set it apart. “Potential” transconjugants were not observed to acquire plasmids, so conjugation did not occur. We would like to discuss the issue of whether these colonies may be false transconjugants–small satellite colonies that may grow around the beta-lactamase-producing colonies as a halo [34]. In the case of *P. aeruginosa*, it is difficult to assume the presence of satellite colonies based on the absence of any carbapenemase-producing colonies and observation of distinct, well-developed large colonies that continue to grow well in medium containing meropenem (Appendix A).

Figure 3 illustrates the fact that, regardless of the donor, the MIC values for *P. aeruginosa* colonies that were selected in mating experiments but did not carry plasmids remain high, ranging between 4 and 8 µg/mL. This observation suggests that *P. aeruginosa* would transform carbapenemase genes or employ additional resistance mechanisms beyond plasmid conjugation and expressing of carbapenemase genes in the presence of meropenem. In fact, the *P. aeruginosa* is characterized by a flexible genome and the ability to implement various antibiotic resistance mechanisms except for carbapenemase production, such as efflux pumps and the modifications in the expression and/or structure of porins [35,36]. However, in our study carbapenemase genes in isolated colonies were not detected using PCR; efflux pump functioning was detected in three of six *P. aeruginosa* isolates. Therefore, in order to gain a better understanding of the complex mechanisms that lead to resistance development in *P. aeruginosa* during their interaction with *K. pneumoniae*, it is essential to expand our research to include a larger number of donor–recipient pairs. Furthermore, a more in-depth genetic analysis is necessary.

Additionally, we would like to discuss our findings regarding the ability of transconjugants to further grow in a medium containing meropenem. As it turns out, not all *E. coli* transconjugant colonies were capable of subsequent growth. In fact, *E. coli* and *P. aeruginosa* “possible” transconjugants did not contain plasmids immediately following the initial cultivation in 75% and 100% of cases, respectively (Appendix A). Slowly growing *E. coli* colonies isolated from matings were unlikely to be satellites, as they did not have a characteristic “satellite” phenotype similar to *P. aeruginosa* (Appendix A), appeared to be separated from each other, and had almost a normal size. However, these colonies were not taken into account when calculating conjugation frequencies, in order to avoid generating inaccurate results. Nevertheless, among these colonies, we identified variants with stable plasmids. When these plasmid-containing variants were cultivated in media with or without meropenem for over 15 passages, they exhibited good growth, retained their plasmids, and were characterized by stable meropenem MICs during periodic assessments (at least 30 times).

The exception was *K. pneumoniae* transconjugants that all acquired the plasmids successfully and replicated them with further cultivation. The recent study on *K. pneumoniae* and *E. coli* demonstrated that, when acquiring the same plasmid from a *K. pneumoniae* donor, the *K. pneumoniae* strain was more conjugation-permissive than the *E. coli* [5,37]. This also applies to our results, where *K. pneumoniae* recipients exhibit 100% plasmid permeability, unlike *E. coli* and, especially, *P. aeruginosa*. The phenomenon of plasmid loss or non-permissiveness, which we assume was inherent in *E. coli* and *P. aeruginosa* isolates, has been previously described and is a common occurrence among bacteria during cell division [38]. Various factors can contribute to plasmid loss in bacterial cells. Several possible explanations for this have been suggested, ranging from a metabolic burden introduced by plasmid replication and the expression of plasmid-encoded genes to disruption of essential host genes due to integration of plasmid DNA, alteration in host gene regulation, and other metabolic consequences, such as the introduction of novel efflux pumps that could potentially remove important biomolecules from the cell [39]. For instance, the host genetic background may play an important role in determining the fitness of plasmids. That is to say, a particular plasmid may decrease fitness in one strain with a specific genetic background, have no noticeable effect on fitness in another, or even be beneficial in yet another strain [38].

It is worth noting that among transconjugant strains with relatively high (2–4 µg/mL) or low (0.25–0.5 µg/mL) meropenem MICs, *E. coli* variants with *bla*_KPC-2_ carbapenemase genes and moderate meropenem MICs equal to 1–2 µg/mL were obtained (Figure 3). These strains have piqued our interest as they are classified as being susceptible to meropenem according to the EUCAST guidelines [22]. It has been recommended that carbapenems be used as monotherapy for treating patients with infections caused by such strains. Therefore, the ability of bacteria to produce carbapenemase enzymes is not taken into account, and susceptibility to meropenem is given the highest priority. Considering this, we decided to investigate how a transconjugant strain that is susceptible to meropenem and produces carbapenemases might behave under meropenem exposure in an in vitro dynamic model (HFIM) that simulates its clinical dosing regimen. Will there be a loss of plasmid, or, conversely, will the ability to resist meropenem improve?

Therefore, in the second part of our study, we conducted a pharmacodynamic evaluation of meropenem using a hollow-fiber infection model and a KPC carbapenemase-producing transconjugant strain *E. coli* C600/565 with a meropenem MIC of 2 µg/mL. We simulated the pharmacokinetic profile observed in epithelial lining fluid following a high dose of meropenem in the hollow-fiber infection model (HFIM), in order to mimic a lung infection scenario. We obtained a series of kinetic time–kill curves that display the course of both the total and resistant bacterial populations for the tested strain. As seen in Figure 5, the ability of an *E. coli* C600/565 strain to produce carbapenemases determines its survival capability during antibiotic exposure as it exhibited extremely intensive growth under meropenem exposure that was not the case with the recipient strain *E. coli* C600. Similar results were obtained in our previous study [40]. To be honest, bacterial resistance to carbapenems in in vitro pharmacodynamic experiments develops slowly or does not develop at all, unless it is caused by the production of carbapenemases [41,42]. Possibly, under the meropenem exposure there appears to be further stabilization of plasmids within cells. This may especially be the case, as the presence of plasmids in cells provides an advantage to their survival. In addition, we assume that intensive growth of resistant cells and the lack of response to meropenem in *E. coli* C600/565 can also be attributed to the upregulation of carbapenemase genes located on the plasmid induced by the antibiotic. This explanation is supported by numerous previous reports of such phenomena in bacteria that are reflected in one of the published studies [31]. Additionally, we observed a similar pattern in the absence of a meropenem effect against the clinical strain of *K. pneumoniae* 1456 producing OXA-48 carbapenemase, which had an MIC of 2 μg/mL, similar to that of *E. coli* C600/565 [43]. In light of the above, it seems that the intensive development of resistance in both instances may be due to the upregulation of carbapenemase genes. Therefore, there is a hidden threat that initially highly susceptible, non-carbapenemase-producing organisms may acquire plasmids carrying carbapenemases genes and may not respond to meropenem therapy, since they can produce carbapenemases. To combat strains for which the effectiveness of meropenem may be significantly reduced, it is essential to explore preventive strategies. The WHO supports countries in reducing antimicrobial resistance by strengthening infection prevention and control measures, such as ensuring effective sanitation and hygiene across all healthcare settings [44]. In addition, a strategy involving shorter durations of antibiotic treatment is relevant at this time, as clinical trials have shown that it minimizes the risk of bacterial resistance to antibiotics [45]. This is due to the fact that, in addition to resistance arising from the production of carbapenemases, clinical strains can also develop porin mutations and activate efflux pumps during prolonged antibiotic treatments [46,47]. In turn, these mechanisms confer cross-resistance to other antibiotics, whether they belong to the same class or not. For example, overexpression of MexAB-OprM confers resistance to quinolones and most beta-lactam antibiotics (including meropenem) [48].

Our study has several limitations. It did not include a large number of strains with a wide range of MICs or other carbapenemase types, for example, metallo-beta-lactamases. Moreover, a more in-depth investigation using genetic methods could also help answer some of the questions raised during the study. Particularly, we did not determine the plasmid copy number in donor strains that is one of the possible factors that may influence their efficacy and therefore conjugation frequency.

## 5. Conclusions

The present study identified several key findings. Firstly, the frequency of conjugation between *K. pneumoniae* and either *K. pneumoniae* or *E. coli* was positively related with meropenem susceptibility of the *K. pneumoniae* donor strain. Secondly, in both recipient strains of *K. pneumoniae*, and *E. coli*, the acquisition of plasmids containing *bla*_KPC_ carbapenemase genes resulted in higher MICs of meropenem compared to those containing *bla*_OXA-48_ carbapenemase genes. Thirdly, the frequency of conjugation between *K. pneumoniae* and *E. coli* falls within similar ranges. Although among the *E. coli* colonies isolated in mating experiments, only approximately 25% accepted plasmids compared to 100% in *K. pneumoniae*. This indicates a higher permissiveness of *K. pneumoniae* for plasmids from donor strains of *K. pneumoniae*, i.e., from bacteria of the same species. Fourthly, conjugation did not occur between *K. pneumoniae* and *P. aeruginosa*, as colonies of *P. aeruginosa* growing on selective agar failed to maintain plasmids. At the same time, their MICs for meropenem increased compared to recipients (4–8 µg/mL vs 0.125–0.25 µg/mL), likely due to alternative resistance mechanisms. A resistance mechanism due to efflux pumps was detected in three of six isolates of *P. aeruginosa*. Finally, the transconjugants with plasmids carrying carbapenemase genes and meropenem MICs at the upper limit of the “susceptibility” range may pose a potential threat to the efficacy of meropenem in clinical settings.

## Figures and Tables

**Figure 1 biomedicines-13-00238-f001:**
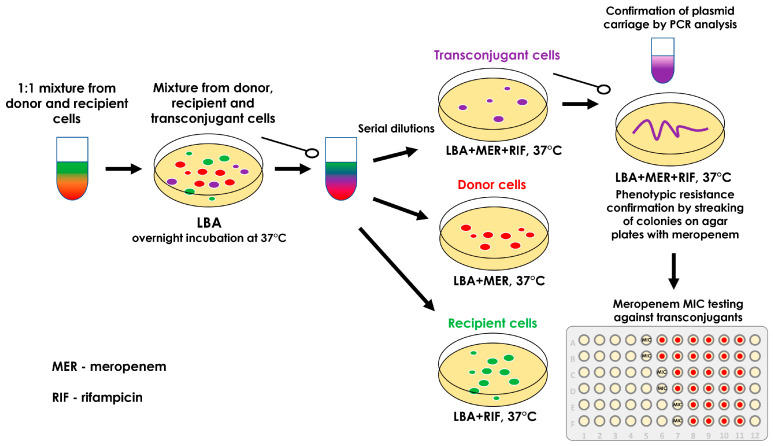
Schematic representation of the conjugation protocol.

**Figure 2 biomedicines-13-00238-f002:**
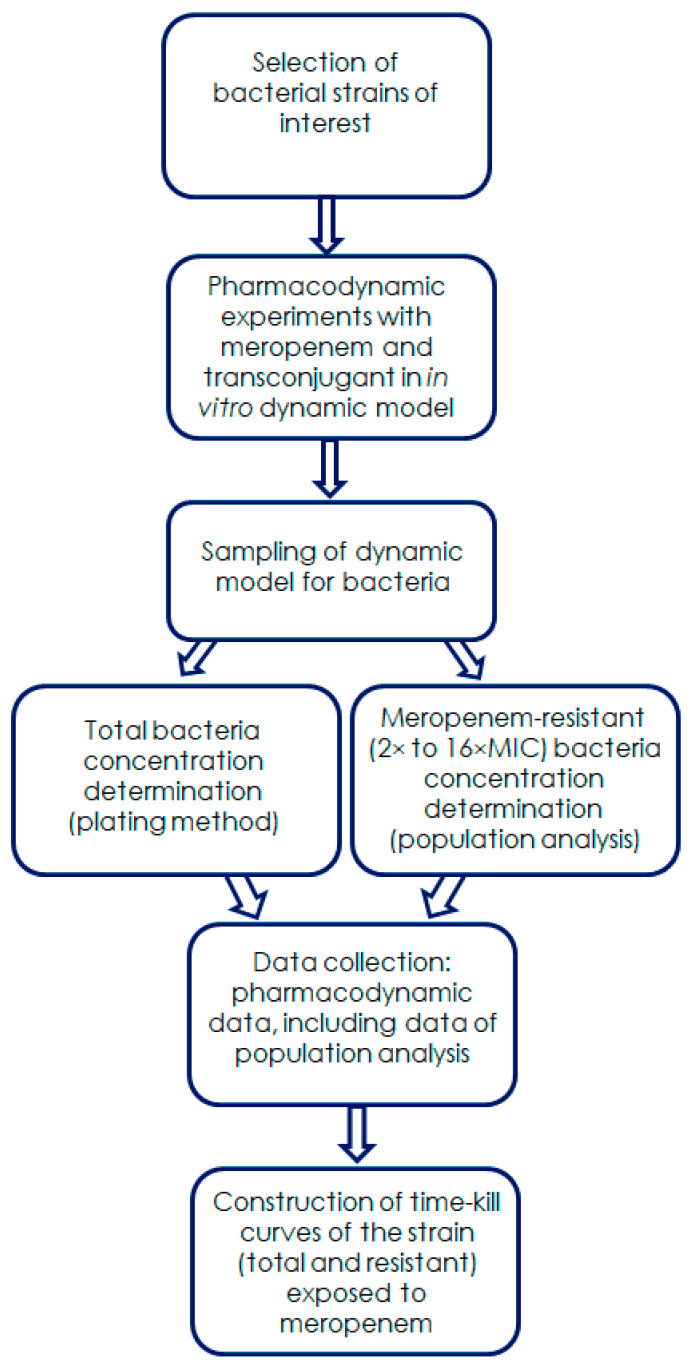
Schematic representation of the pharmacodynamic experiment protocol.

**Figure 3 biomedicines-13-00238-f003:**
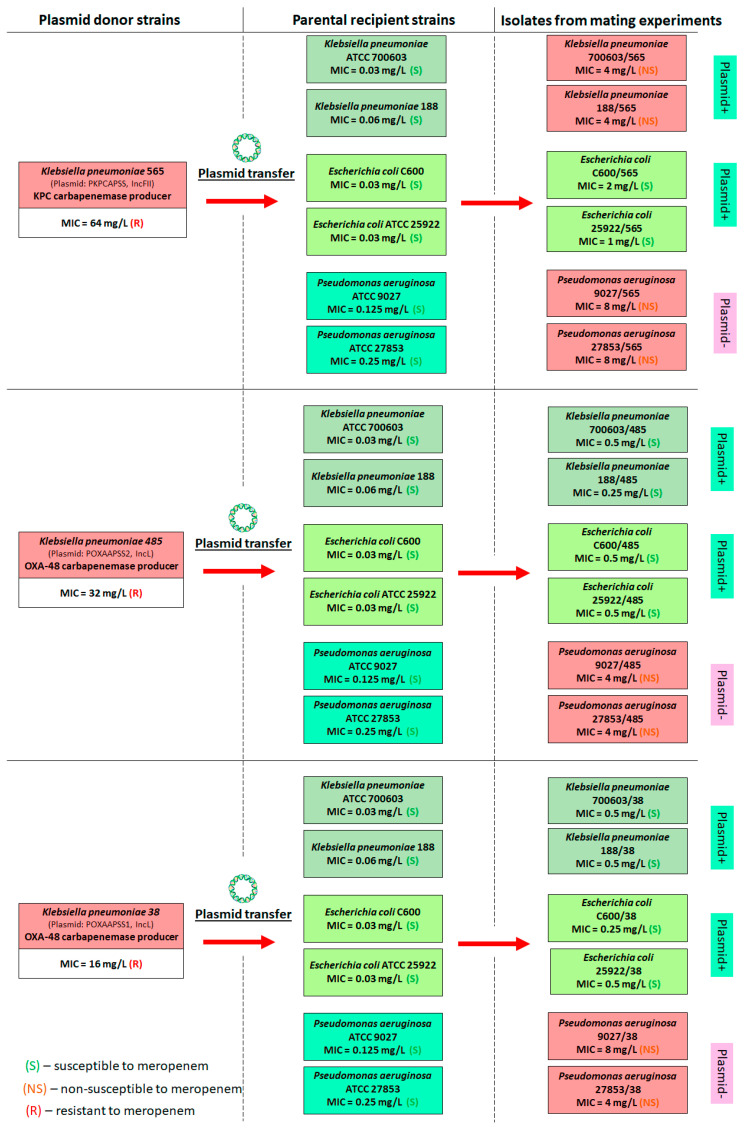
Flowchart followed to perform bacterial mating in pairs *K. pneumoniae*–*K. pneumoniae*, *K. pneumoniae*–*E. coli*, and *K. pneumoniae*–*P. aeruginosa*, and meropenem susceptibility of transconjugant strains. “Plasmid+” or “plasmid−” indicates the confirmation or not of plasmids in isolates that were tested for meropenem susceptibility. Susceptibility breakpoints were based on EUCAST recommendations: ≤2 µg/mL—susceptible, >8 µg/mL—resistant [22].

**Figure 4 biomedicines-13-00238-f004:**
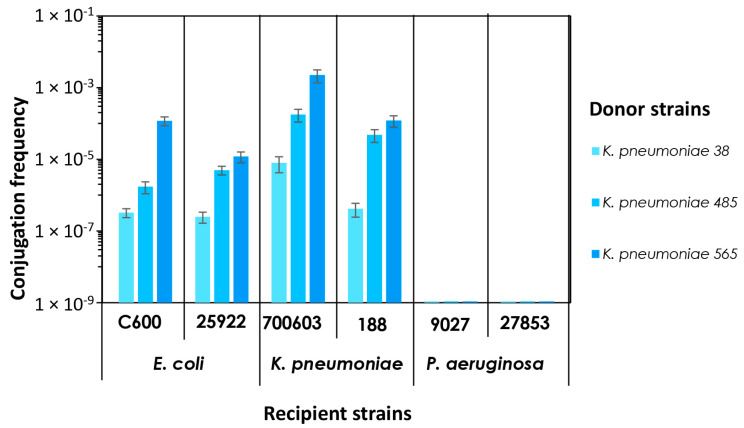
The PCR-based conjugation frequency between donor carbapenemase-producing *K. pneumoniae* strains and plasmid-free recipient strains of *K. pneumoniae*, *E. coli*, and *P. aeruginosa*. The data presented as arithmetic means ± standard deviations (n = 3).

**Figure 5 biomedicines-13-00238-f005:**
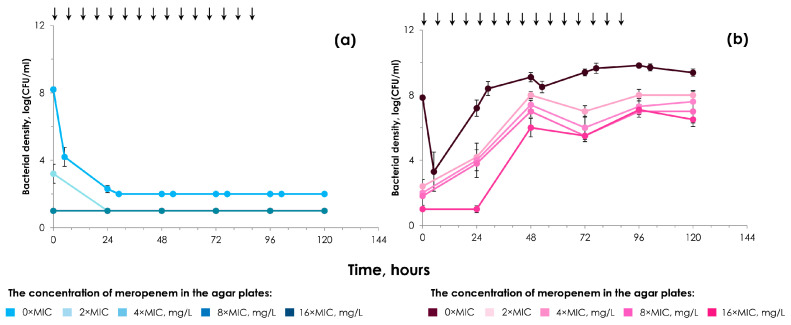
Time courses of the total bacterial population (0 × MIC) and meropenem-resistant (2×, 4×, 8× and 16 × MIC) sub-populations of recipient (**a**) and transconjugant (**b**) carbapenemase-producing strain of *E. coli* in pharmacodynamic experiments. Arrows indicate the start of the meropenem infusion. The data presented as arithmetic means± standard deviation (n = 3). Error bars represent standard deviation.

**Table 1 biomedicines-13-00238-t001:** General characteristics of donor *K. pneumoniae* strains and their plasmids.

*K. pneumoniae* Strain Number	38	485	565
City, year	Moscow, 2011	Saint Petersburg, 2012	Saint Petersburg, 2012
Plasmid	pOXAAPSS2	pOXAAPSS1	pKPCAPSS
NCBI reference sequence	NZ_KU159086.1	NZ_KU159085.1	NZ_KP008371.1
Incompatibility group	IncL	IncL	IncFII
Length	63,359 bp	66,284 bp	127,970 bp
Resistome	*bla* _OXA-48_	*bla*_OXA-48_, *bla*_TEM-1b_	*bla*_KPC-2_, *bla*_TEM-1b_, *qnrS1*, *mphA*, *mrx*, *mphR*
ST of donor	ST147	ST395	ST273
MIC of meropenem, µg/mL	16	32	64

ST—sequence type; MIC—minimum inhibitory concentration.

**Table 2 biomedicines-13-00238-t002:** MICs of meropenem against tested bacterial strains.

Bacterial Strain	Carbapenemase	Meropenem MIC, µg/mL
*Klebsiella pneumoniae* 38	OXA-48	16
*Klebsiella pneumoniae* 485	OXA-48	32
*Klebsiella pneumoniae* 565	KPC	64
*Klebsiella pneumoniae* ATCC 700603 ^1^	None	0.06
*Klebsiella pneumoniae* 188 ^1^	None	0.03
*Escherichia coli* ATCC 25922 ^1^	None	0.03
*Escherichia coli* C600 ^1^	None	0.03
*Pseudomonas aeruginosa* ATCC 9027 ^1^	None	0.125
*Pseudomonas aeruginosa* ATCC 27853 ^1^	None	0.25

^1^—rifampicin-resistant mutants.

**Table 3 biomedicines-13-00238-t003:** MICs of meropenem in the presence and absence of efflux pump inhibitor CCCP against tested *P. aeruginosa* isolates from matings with *K. pneumoniae*.

*P. aeruginosa* Isolate	Meropenem MIC, µg/mL	Meropenem MIC in the Presence of CCCP, µg/mL
565/9027	16	16
**565/27853**	**4**	**0.125**
485/9027	16	8
**485/27853**	**16**	**1**
**38/9027**	**4**	**0.125**
38/27853	8	8

In **bold** are highlighted *P. aeruginosa* strains, for which efflux has been confirmed through a 4-fold or higher decrease in meropenem MICs in the presence of CCCP.

## Data Availability

The original contributions presented in this study are included in the article/Appendix A. Further inquiries can be directed to the corresponding author.

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
