# Peer review of "Intra- and Interspecies Conjugal Transfer of Plasmids in Gram-Negative Bacteria"

_biomedicines, 2025, doi:10.3390/biomedicines13010238_

Round 1

Reviewer 1 Report

Comments and Suggestions for Authors

I appreciate the efforts of Savelieva et al. to incorporate the suggestions and corrections in the manuscript. I commend the authors for their work. I endorse publication of the manuscript in its current form. 

Reviewer 2 Report

Comments and Suggestions for Authors

The authors have conducted an interesting study with the aim of investigating the frequency of transfer of carbapenemase-encoding plasmids from K. pneumoniae to E. coli and P. aeruginosa. This is an intriguing manuscript that adds significant information to the existing literature, given the increasing prevalence of multidrug-resistant infections in hospital and nosocomial settings, and particularly the growing and severe resistance to carbapenems. Below are my comments.

Line 45 - Add the main infections that can be caused by Klebsiella.

Line 75 - In Table 1, include a legend with the abbreviations spelled out.

The materials and methods are appropriate for the purpose of the manuscript. The statistical analysis is consistent with the objective of the paper, as is the calculation of the MIC.

Line 335 - Meropenem resistance and carbapenem resistance in general can be associated with several other cross-resistances; for the convenience of readers, it would be helpful to specify these in the manuscript. This can be done in the discussion or results section.

Line 427 - Given the increasing prevalence of multidrug-resistant infections and the difficulty in finding targeted antibiotic therapies, what is the authors’ perspective on possible preventive strategies? Recently, several studies have highlighted how targeted antibiotic therapies of reduced duration reduce the incidence of subsequent multidrug-resistant infections. These findings are well validated in the pediatric population (see 10.3390/children9111647) with promising results also in the adult population. It would be useful to elaborate on this concept in the discussion.

Minor improvements in the English language translation are necessary.

Round 2

Reviewer 2 Report

Comments and Suggestions for Authors

The authors have adequately addressed all my comments. The manuscript has significantly improved. Given the importance of the topic of antibiotic resistance, I believe the paper can be accepted in its current form, subject to the Editor's final decision.